# Fast Sampling-Based Inference in Balanced Neuronal Networks

**Guillaume Hennequin**[1]
gjeh2@cam.ac.uk

**Laurence Aitchison**[2]
laurence@gatsby.ucl.ac.uk

**Máté Lengyel**[1]
m.lengyel@eng.cam.ac.uk

[1]Computational & Biological Learning Lab, Dept. of Engineering, University of Cambridge, UK
[2]Gatsby Computational Neuroscience Unit, University College London, UK

## Abstract

Multiple lines of evidence support the notion that the brain performs probabilistic inference in multiple cognitive domains, including perception and decision making. There is also evidence that probabilistic inference may be implemented in the brain through the (quasi-)stochastic activity of neural circuits, producing samples from the appropriate posterior distributions, effectively implementing a Markov chain Monte Carlo algorithm. However, time becomes a fundamental bottleneck in such sampling-based probabilistic representations: the quality of inferences depends on how fast the neural circuit generates new, uncorrelated samples from its stationary distribution (the posterior). We explore this bottleneck in a simple, linear-Gaussian latent variable model, in which posterior sampling can be achieved by stochastic neural networks with linear dynamics. The well-known Langevin sampling (LS) recipe, so far the only sampling algorithm for continuous variables of which a neural implementation has been suggested, naturally fits into this dynamical framework. However, we first show analytically and through simulations that the symmetry of the synaptic weight matrix implied by LS yields critically slow mixing when the posterior is high-dimensional. Next, using methods from control theory, we construct and inspect networks that are optimally fast, and hence orders of magnitude faster than LS, while being far more biologically plausible. In these networks, strong – but transient – selective amplification of external noise generates the spatially correlated activity fluctuations prescribed by the posterior. Intriguingly, although a detailed balance of excitation and inhibition is dynamically maintained, detailed balance of Markov chain steps in the resulting sampler is violated, consistent with recent findings on how statistical irreversibility can overcome the speed limitation of random walks in other domains.

## 1 Introduction

The high speed of human sensory perception [1] is puzzling given its inherent computational complexity: sensory inputs are noisy and ambiguous, and therefore do not uniquely determine the state of the environment for the observer, which makes perception akin to a statistical inference problem. Thus, the brain must represent and compute with complex and often high-dimensional probability distributions over relevant environmental variables. Most state-of-the-art machine learning techniques for large scale inference trade inference accuracy for computing speed (e.g. [2]). The brain, on the contrary, seems to enjoy both simultaneously [3].

Some probabilistic computations can be made easier through an appropriate choice of representation for the probability distributions of interest. Sampling-based representations used in Monte Carlo

techniques, for example, make computing moments of the distribution or its marginals straightforward. Indeed, recent behavioural and neurophysiological evidence suggests that the brain uses such sampling-based representations by neural circuit dynamics implementing a Markov chain Monte Carlo (MCMC) algorithm such that their trajectories in state space produce sequential samples from the appropriate posterior distribution [4, 5, 6].

However, for sampling-based representations, speed becomes a key bottleneck: computations involving the posterior distribution become accurate only after enough samples have been collected, and one has no choice but *to wait* for those samples to be delivered by the circuit dynamics. For sampling to be of any practical use, the interval that separates the generation of two independent samples must be short relative to the desired behavioral timescale. Single neurons can integrate their inputs on a timescale $\tau_{\mathrm{m}} \approx 10 - 50$ ms, whereas we must often make decisions in less than a second: this leaves just enough time to use (i.e. read out) a few tens of samples. What kinds of neural circuit dynamics are capable of producing *uncorrelated* samples at $\sim$100 Hz remains unclear.

Here, we introduce a simple yet non-trivial generative model and seek plausible neuronal network dynamics for *fast* sampling from the corresponding posterior distribution. While some standard machine learning techniques such as Langevin or Gibbs sampling do suggest "neural network"-type solutions to sampling, not only are the corresponding architectures implausible in fundamental ways (e.g. they violate Dale's law), but we show here that they lead to unacceptably slow mixing in high dimensions. Although the issue of sampling speed in general is well appreciated in the context of machine learning, there have been no systematic approaches to tackle it owing to a large part to the fact that sampling speed can only be evaluated empirically in most cases. In contrast, the simplicity of our generative model allowed us to draw an analytical picture of the problem which in turn suggested a systematic approach for solving it. Specifically, we used methods from robust control to discover the *fastest* neural-like sampler for our generative model, and to study its structure. We find that it corresponds to greatly non-symmetric synaptic interactions (leading to statistical irreversibility), and mathematically nonnormal[1] circuit dynamics [7, 8] in close analogy with the dynamical regime in which the cortex has been suggested to operate [9].

## 2   Linear networks perform sampling under a linear Gaussian model

We focus on a linear Gaussian latent variable model which generates observations $\mathbf{h} \in \mathbb{R}^M$ as weighted sums of $N$ features $\mathbf{A} \equiv (\mathbf{a}_1; \ldots; \mathbf{a}_N) \in \mathbb{R}^{M \times N}$ with jointly Gaussian coefficients $\mathbf{r} \in \mathbb{R}^N$, plus independent additive noise terms (Fig. 1, left). More formally:

$$p(\mathbf{r}) = \mathcal{N}(\mathbf{r}; 0, \mathbf{C}) \qquad \text{and} \qquad p(\mathbf{h}|\mathbf{r}) = \mathcal{N}\left(\mathbf{h}; \mathbf{Ar}, \sigma_h^2 \mathbf{I}\right) \tag{1}$$

where $\mathbf{I}$ denotes the identity matrix. The posterior distribution is multivariate Gaussian, $p(\mathbf{r}|\mathbf{h}) = \mathcal{N}\left(\mathbf{r}; \boldsymbol{\mu}(\mathbf{h}), \boldsymbol{\Sigma}\right)$, with

$$\boldsymbol{\Sigma} = \left(\mathbf{C}^{-1} + \mathbf{A}^\top \mathbf{A}/\sigma_h^2\right)^{-1} \qquad \text{and} \qquad \boldsymbol{\mu}(\mathbf{h}) = \boldsymbol{\Sigma}\mathbf{A}^\top \mathbf{h}/\sigma_h^2. \tag{2}$$

where we made explicit the fact that under this simple model, only the mean, $\boldsymbol{\mu}(\mathbf{h})$, but not the covariance of the posterior, $\boldsymbol{\Sigma}$, depends on the input, $\mathbf{h}$.

We are interested in neural circuit dynamics for sampling from $p(\mathbf{r}|\mathbf{h})$, whereby the data (observation) $\mathbf{h}$ is given as a constant feedforward input to a population of recurrently connected neurons, each of which encodes one of the latent variables and also receives inputs from an external, private source of noise $\boldsymbol{\xi}$ (Fig. 1, right). Our goal is to devise a network such that the activity fluctuations $\mathbf{r}(t)$ in the recurrent layer have a stationary distribution that matches the posterior, *for any* $\mathbf{h}$.

Specifically, we consider linear recurrent stochastic dynamics of the form:

$$d\mathbf{r} = \frac{dt}{\tau_{\mathrm{m}}}\left[-\mathbf{r}(t) + \mathbf{Wr}(t) + \mathbf{Fh}\right] + \sigma_\xi \sqrt{\frac{2}{\tau_{\mathrm{m}}}}\, d\boldsymbol{\xi}(t) \tag{3}$$

where $\tau_{\mathrm{m}} = 20$ ms is the single-unit "membrane" time constant, and $d\boldsymbol{\xi}$ is a Wiener process of unit variance, which is scaled by a scalar noise intensity $\sigma_\xi$. The activity $r_i(t)$ could represent either the

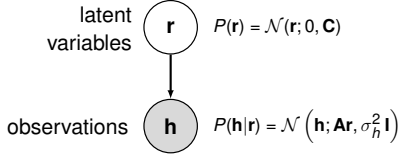

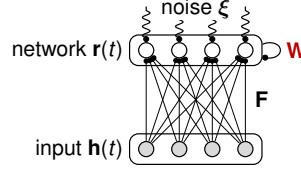

Figure 1: **Sampling under a linear Gaussian latent variable model using neuronal network dynamics.** Left: schematics of the generative model. Right: schematics of the recognition model. See text for details.

membrane potential of neuron $i$, or the deviation of its momentary firing rate from a baseline. The matrices $\mathbf{F}$ and $\mathbf{W}$ contain the feedforward and recurrent connection weights, respectively.

The stationary distribution of $\mathbf{r}$ is indeed Gaussian with a mean $\boldsymbol{\mu}^{\mathbf{r}}(\mathbf{h}) = (\mathbf{I} - \mathbf{W})^{-1}\mathbf{Fh}$ and a covariance matrix $\boldsymbol{\Sigma}^{\mathbf{r}} \equiv \left\langle (\mathbf{r}(t) - \boldsymbol{\mu}^{\mathbf{r}})(\mathbf{r}(t) - \boldsymbol{\mu}^{\mathbf{r}})^{\top} \right\rangle_t$. For the following, we will use the dependence of $\boldsymbol{\Sigma}^{\mathbf{r}}$ on $\mathbf{W}$ (and $\sigma_{\xi}$) given implicitly by the following Lyapunov equation [10]:

$$(\mathbf{W} - \mathbf{I})\boldsymbol{\Sigma}^{\mathbf{r}} + \boldsymbol{\Sigma}^{\mathbf{r}}(\mathbf{W} - \mathbf{I})^{\top} = -2\sigma_{\xi}^2\mathbf{I} \tag{4}$$

Note that in the absence of recurrent connectivity ($\mathbf{W} = 0$), the variance of every $r_i(t)$ would be exactly $\sigma_{\xi}^2$. Note also that, just as required (see above), only the mean, $\boldsymbol{\mu}^{\mathbf{r}}(\mathbf{h})$, but not the covariance, $\boldsymbol{\Sigma}^{\mathbf{r}}$, depends on the input, $\mathbf{h}$.

In order for the dynamics of Eq. 3 to sample from the correct posteriors, we must choose $\mathbf{F}$, $\mathbf{W}$ and $\sigma_{\xi}$ such that $\boldsymbol{\mu}^{\mathbf{r}}(\mathbf{h}) = \boldsymbol{\mu}(\mathbf{h})$ for any $\mathbf{h}$, and $\boldsymbol{\Sigma}^{\mathbf{r}} = \boldsymbol{\Sigma}$. One possible solution (which, importantly, is not unique, as we show later) is

$$\mathbf{F} = (\sigma_{\xi}/\sigma_h)^2 \ \mathbf{A}^{\top} \qquad \text{and} \qquad \mathbf{W} = \mathbf{W}_{\mathrm{L}} \equiv \mathbf{I} - \sigma_{\xi}^2 \ \boldsymbol{\Sigma}^{-1} \tag{5}$$

with arbitrary $\sigma_{\xi} > 0$.

In the following, we will be interested in the likelihood matrix $\mathbf{A}$ only insofar as it affects the posterior covariance matrix $\boldsymbol{\Sigma}$, which turns out to be the main determinant of sampling speed. We will therefore directly choose some covariance matrix $\boldsymbol{\Sigma}$, and set $\mathbf{h} = 0$ without loss of generality.

## 3 Langevin sampling is very slow

Langevin sampling (LS) is a common sampling technique [2, 11, 12], and in fact the only one that has been proposed to be neurally implemented for continuous variables [6, 13]. According to LS, a stochastic dynamical system performs "noisy gradient ascent of the log posterior":

$$\mathrm{d}\mathbf{r} = \frac{1}{2}\frac{\partial}{\partial \mathbf{r}} \log \ p(\mathbf{r}|\mathbf{h}) \, \mathrm{d}t + \mathrm{d}\boldsymbol{\xi} \tag{6}$$

where $\mathrm{d}\boldsymbol{\xi}$ is a unitary Wiener process. When $\mathbf{r}|\mathbf{h}$ is Gaussian, Eq. 6 reduces to Eq. 3 for $\sigma_{\xi} = 1$ and the choice of $\mathbf{F}$ and $\mathbf{W}$ given in Eq. 5 – hence the notation $\mathbf{W}_{\mathrm{L}}$ above. Note that $\mathbf{W}_{\mathrm{L}}$ is symmetric.

As we show now, this choice of weight matrix leads to critically slow mixing (i.e. very long autocorrelation time scales in $\mathbf{r}(t)$) when $N$ is large. In a linear network, the average autocorrelation length is dominated by the decay time constant $\tau_{\max}$ of the slowest eigenmode, i.e. the eigenvector of $(\mathbf{W} - \mathbf{I})$ associated with the eigenvalue $\lambda_{\max}^{\mathbf{W}-\mathbf{I}}$ which, of all the eigenvalues of $(\mathbf{W} - \mathbf{I})$, has the largest real part (which must still be negative, to ensure stability). The contribution of the slowest eigenmode to the sample autocorrelation time is $\tau_{\max} = -\tau_{\mathrm{m}}/\mathrm{Re}\left(\lambda_{\max}^{\mathbf{W}-\mathbf{I}}\right)$, so sampling becomes very slow when $\mathrm{Re}\left(\lambda_{\max}^{\mathbf{W}-\mathbf{I}}\right)$ approaches 0. This is, in fact, what happens with LS as $N \to \infty$. Indeed, we could derive the following generic lower bound (details can be found in our Supplementary Information, SI):

$$\lambda_{\max}^{\mathbf{W}_{\mathrm{L}}-\mathbf{I}} \quad \geq \quad \frac{-(\sigma_{\xi}/\sigma_0)^2}{\sqrt{1 + N\sigma_r^2}} \tag{7}$$

which is shown as dashed lines in Fig. 2. Thus, LS becomes infinitely slow in the large $N$ limit when pairwise correlations do not vanish in that limit (or at least not as fast as $N^{-\frac{1}{2}}$ in their std.).

Slowing becomes even worse when $\boldsymbol{\Sigma}$ is drawn from the inverse Wishart distribution with $\nu$ degrees of freedom and scale matrix $\omega^{-2}\mathbf{I}$ (Fig. 2). We choose $\nu = N - 1 + \lfloor\sigma_r^{-2}\rfloor$ and $\omega^{-2} = \sigma_0^2(\nu - N - 1)$

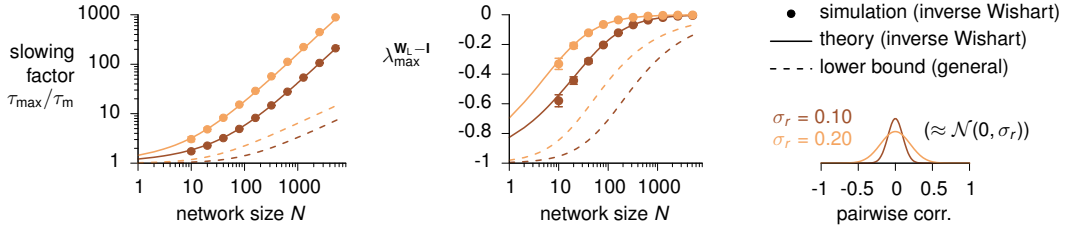

Figure 2: **Langevin sampling (LS) is slow in high-dimension**. Random covariance matrices $\boldsymbol{\Sigma}$ of size $N$ are drawn from an inverse Wishart distribution with parameters chosen such that the average diagonal element (variance) is $\sigma_0^2 = 1$ and the distribution of pairwise correlations has zero mean and variance $\sigma_r^2$ (right). Sampling from $\mathcal{N}(\mathbf{0}, \boldsymbol{\Sigma})$ using a stochastic neural network (cf. Fig. 1) with $\mathbf{W} = \mathbf{W}_{\mathrm{L}}$ (LS, symmetric solution) becomes increasingly slow as $N$ grows, as indicated by the relative decay time constant $\tau_{\max}/\tau_{\mathrm{m}}$ of the slowest eigenmode of $(\mathbf{W}_{\mathrm{L}} - \mathbf{I})$ (left), which is also the negative inverse of its largest eigenvalue (middle). Dots indicate the numerical evaluation of the corresponding quantities, and errorbars (barely noticeable) denote standard deviation across several random realizations of $\boldsymbol{\Sigma}$. Dashed lines correspond to the generic bound in Eq. 7. Solid lines are obtained from random matrix theory under the asssumption that $\boldsymbol{\Sigma}$ is drawn from an inverse Wishart distribution (Eq. 8). Parameters: $\sigma_\xi = \sigma_0 = 1$.

such that the expected value of a diagonal element (variance) in $\boldsymbol{\Sigma}$ is $\sigma_0^2$, and the distribution of pairwise correlations is centered on zero with variance $\sigma_r^2$. The asymptotic behavior of the largest eigenvalue of $\boldsymbol{\Sigma}^{-1}$ (the square of the smallest singular value of a random $\nu \times N$ rectangular matrix) is known from random matrix theory (e.g. [14]), and we have for large $N$:

$$\lambda_{\max}^{\mathbf{W}_{\mathrm{L}}-\mathbf{I}} \approx -\frac{(\sigma_\xi/\sigma_0)^2}{\lfloor \sigma_r^{-2} \rfloor - 2} \left( \sqrt{N - 1 + \lfloor \sigma_r^{-2} \rfloor} - \sqrt{N} \right)^2 \qquad \sim -\mathcal{O}\left(\frac{1}{N}\right) \qquad (8)$$

This scaling behavior is shown in Fig. 2 (solid lines). In fact, we can also show (cf. SI) that LS is (locally) the slowest possible choice (see Sec. 4 below for a precise definition of "slowest", and SI for details).

Note that both Eqs. 7-8 are inversely proportional to the ratio $(\sigma_0/\sigma_\xi)$, which tells us how much the recurrent interactions must amplify the external noise in order to produce samples from the right stationary activity distribution. The more amplification is required ($\sigma_0 \gg \sigma_\xi$), the slower the dynamics of LS. Conversely, one could potentially make Langevin sampling faster by increasing $\sigma_\xi$, but $\sigma_\xi$ would need to scale as $\sqrt{N}$ to annihilate the critical slowing problem. This – in itself – is unrealistic; moreover, it would also require the resulting connectivity matrix to have a large negative diagonal ($\mathcal{O}(-N)$) – ie. the intrinsic neuronal time constant $\tau_{\mathrm{m}}$ to scale as $\mathcal{O}(1/N)$ –, which is perhaps even more unrealistic.[2]

Note also that LS can be sped up by appropriate "preconditioning" (e.g. [15, 16]), for example using the inverse Hessian of the log-posterior. In our case, a simple calculation shows that this corresponds to removing all recurrent connections, and pushing the posterior covariance matrix to the external noise sources, which is only postponing the problem to some other brain network.

Finally, LS is fundamentally implausible as a neuronal implementation: it imposes symmetric synaptic interactions, which is simply not possible in the brain due to the existence of distinct classes of excitatory and inhibitory neurons (Dale's principle). In the following section, we show that networks can be constructed that overcome all the above limitations of LS in a principled way.

## 4 General solution and quantification of sampling speed

While Langevin dynamics (Eq. 6) provide a general recipe for sampling from any given posterior density, they unduly constrain the recurrent interactions to be symmetric – at least in the Gaussian

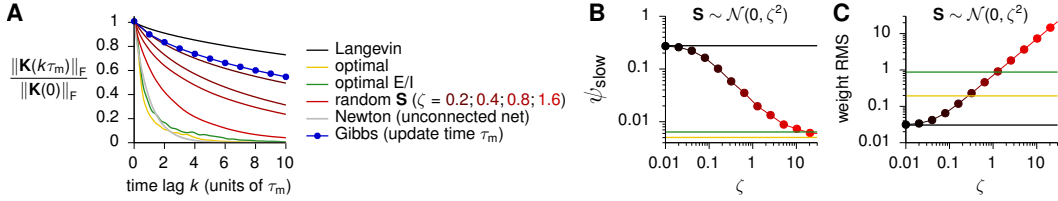

Figure 3: **How fast is the fastest sampler?** (**A**) Scalar measure of the statistical dependency between any two samples collected $k\tau_\mathrm{m}$ seconds apart (cf. main text), for Langevin sampling (black), Gibbs sampling (blue, assuming a full update sweep is done every $\tau_\mathrm{m}$), a series of networks (brown to red) with connectivities given by Eq. 9 where the elements of the skew-symmetric matrix $\mathbf{S}$ were drawn iid. from $\mathcal{N}(0, \zeta^2)$ for different values of $\zeta$ (see also panel B), the unconstrained optimized network (yellow), and the optimized E/I network (green). For reference, the dashed gray line shows the behavior of a network in which there are no recurrent interactions, and the posterior covariance is encoded in the covariance of the input noise, which in fact corresponds to Langevin sampling with inverse Hessian ("Newton"-like) preconditioning [16]. (**B**) Total slowing cost $\psi_\mathrm{slow}(\mathbf{S})$ when $S_{i<j} \sim \mathcal{N}(0, \zeta^2)$, for increasing values of $\zeta$. The Langevin and the two optimized networks are shown as horizontal lines for comparison. (**C**) Same as in (B), showing the root mean square (RMS) value of the synaptic weights. Parameter values: $N = 200$, $N_I = 100$, $\sigma_\xi = 1$, $\tau_\mathrm{m} = 20$ ms.

case. To see why this is a drastic restriction, let us observe that any connectivity matrix of the form

$$\mathbf{W}(\mathbf{S}) = \mathbf{I} + \left(-\sigma_\xi^2 \mathbf{I} + \mathbf{S}\right) \mathbf{\Sigma}^{-1} \qquad (9)$$

where $\mathbf{S}$ is an arbitrary skew-symmetric matrix ($\mathbf{S}^\top = -\mathbf{S}$), solves Eq. 4, and therefore induces the correct stationary distribution $\mathcal{N}(\cdot, \mathbf{\Sigma})$ under the linear stochastic dynamics of Eq. 3. Note that Langevin sampling corresponds to $\mathbf{S} = 0$ (cf. Eq. 5). In general, though, there are $\mathcal{O}(N^2)$ degrees of freedom in the skew-symmetric matrix $\mathbf{S}$, which could perhaps be exploited to increase the mixing rate. In Sec. 5, we will show that indeed a large gain in sampling speed can be obtained through an appropriate choice of $\mathbf{S}$. For now, let us quantify slowness.

Let $\mathbf{\Lambda} \equiv \mathrm{diag}\left(\mathbf{\Sigma}\right)$ be the diagonal matrix that contains all the posterior variances, and $\mathbf{K}(\mathbf{S}, \tau) \equiv \left\langle (\mathbf{r}(t+\tau) - \mu)(\mathbf{r}(t) - \mu)^\top \right\rangle_t$ be the matrix of lagged covariances among neurons under the stationary distribution of the dynamics (so that $\mathbf{\Lambda}^{-\frac{1}{2}} \mathbf{K}(\mathbf{S}, \tau) \mathbf{\Lambda}^{-\frac{1}{2}}$ is the autocorrelation matrix of the network). Note that $\mathbf{K}(\mathbf{S}, 0) = \mathbf{\Sigma}$ is the posterior covariance matrix, and that for fixed $\mathbf{\Sigma}$, $\sigma_\xi^2$ and $\tau_\mathrm{m}$, $\mathbf{K}(\mathbf{S}, \tau)$ depends only on the lag $\tau$ and on the matrix of recurrent weights $\mathbf{W}$, which itself depends only on the skew-symmetric matrix $\mathbf{S}$ of free parameters. We then define a "total slowing cost"

$$\psi_\mathrm{slow}(\mathbf{S}) = \frac{1}{2\tau_\mathrm{m} N^2} \int_0^\infty \left\| \mathbf{\Lambda}^{-\frac{1}{2}} \mathbf{K}(\mathbf{S}, \tau) \mathbf{\Lambda}^{-\frac{1}{2}} \right\|_\mathrm{F}^2 \mathrm{d}\tau \qquad (10)$$

which penalizes the magnitude of the temporal (normalized) autocorrelations and pairwise cross-correlations in the sequence of samples generated by the circuit dynamics. Here $\|\mathbf{M}\|_F^2 \equiv \mathrm{trace}(\mathbf{M}\mathbf{M}^\top) = \sum_{ij} M_{ij}^2$ is the squared Frobenius norm of $\mathbf{M}$.

Using the above measure of slowness, we revisit the mixing behavior of LS on a toy covariance matrix $\mathbf{\Sigma}$ drawn from the same inverse Wishart distribution mentioned above with parameters $N = 200$, $\sigma_0^2 = 2$ and $\sigma_r = 0.2$. We further regularize $\mathbf{\Sigma}$ by adding the identity matrix to it, which does not change anything in terms of the scaling law of Eq. 8 but ensures that the diagonal of $\mathbf{W}_\mathrm{L}$ remains bounded as $N$ grows large. We will use the same $\mathbf{\Sigma}$ in the rest of the paper. Figure 3A shows $\left\| \mathbf{\Lambda}^{-1/2} \mathbf{K}(\mathbf{S}, \tau) \mathbf{\Lambda}^{-1/2} \right\|_\mathrm{F}$ as a function of the time lag $\tau$: as predicted in Sec. 3, mixing is indeed an order of magnitude slower for LS ($\mathbf{S} = \mathbf{0}$, solid black line) than the single-neuron time constant $\tau_\mathrm{m}$ (grey dashed line). Note that $\psi_\mathrm{slow}$ (Eq. 10, Fig. 3B) is proportional to the area under the squared curve shown in Fig. 3A. Sample activity traces for this network, implementing LS, can be found in Fig. 4B (top).

Using the same measure of slowness, we also inspected the speed of Gibbs sampling, another widely used sampling technique (e.g. [17]) inspiring neural network dynamics for sampling from distributions over binary variables [18, 19, 20]. Gibbs sampling defines a Markov chain that operates in

discrete time, and also uses a symmetric weight matrix. In order to compare its mixing speed with that of our continuous stochastic dynamics, we assume that a full update step (in which all neurons have been updated once) takes time $\tau_{\mathrm{m}}$. We estimated the integrand of the slowing cost (Eq. 10) numerically using 30'000 samples generated by the Gibbs chain (Fig. 3A, blue). Gibbs sampling is comparable to LS here: samples are still correlated on a timescale of order $\sim 50\,\tau_{\mathrm{m}}$.

Finally, one may wonder how a random choice of $\mathbf{S}$ would perform in terms of decorrelation speed. We drew random skew-symmetric $\mathbf{S}$ matrices from the Gaussian ensemble, $S_{i<j} \sim \mathcal{N}(0, \zeta^2)$, and computed the slowing cost (Fig. 3, red). As the magnitude $\zeta$ of $\mathbf{S}$ increases, sampling becomes faster and faster until the dynamics is about as fast as the single-neuron time constant $\tau_{\mathrm{m}}$. However, the synaptic weights also grow with $\zeta$ (Fig. 3C), and we show in Sec. 5 that an even faster sampler exists that has comparatively weaker synapses. It is also interesting to note that the slope of $\psi_{\mathrm{slow}}$ at $\zeta = 0$ is zero, suggesting that LS is in fact maximally slow (we prove this formally in the SI).

## 5    What is the fastest sampler?

We now show that the skew-symmetric matrix $\mathbf{S}$ can be optimized for sampling speed, by directly minimizing the slowing cost $\psi_{\mathrm{slow}}(\mathbf{S})$ (Eq. 10), subject to an $L_2$-norm penalty. We thus seek to minimize:

$$\mathcal{L}(\mathbf{S}) \quad \equiv \quad \psi_{\mathrm{slow}}(\mathbf{S}) + \frac{\lambda_{L_2}}{2N^2} \left\| \mathbf{W}(\mathbf{S}) \right\|_F^2 . \tag{11}$$

The key to performing this minimization is to use classical Ornstein-Uhlenbeck theory (e.g. [10]) to bring our slowness cost under a form mathematically analogous to a different optimization problem that has arisen recently in the field of robust control [21]. We can then use analytical results obtained there concerning the gradient of $\psi_{\mathrm{slow}}$, and obtain the overall gradient:

$$\frac{\partial \mathcal{L}(\mathbf{S})}{\partial \mathbf{S}} \quad = \quad \frac{1}{N^2} \left[ (\mathbf{\Sigma}^{-1}\mathbf{P}\mathbf{Q})^{\top} - (\mathbf{\Sigma}^{-1}\mathbf{P}\mathbf{Q}) \right] + \frac{\lambda_{L_2}}{N^2} \left[ \mathbf{S}\mathbf{\Sigma}^{-2} + \mathbf{\Sigma}^{-2}\mathbf{S} \right] \tag{12}$$

where matrices $\mathbf{P}$ and $\mathbf{Q}$ are obtained by solving two dual Lyapunov equations. All details can be found in our SI.

We initialized $\mathbf{S}$ with random, weak and uncorrelated elements (cf. the end of Sec. 4, with $\zeta = 0.01$), and ran the L-BFGS optimization algorithm using the gradient of Eq. 12 to minimize $\mathcal{L}(\mathbf{S})$ (with $\lambda_{L_2} = 0.1$). The resulting, optimal sampler is an order of magnitude faster than either Langevin or Gibbs sampling: samples are decorrelated on a timescale that is even faster than the single-neuron time constant $\tau_{\mathrm{m}}$ (Fig. 3A, orange). We also found that fast solutions (with correlation length $\sim \tau_m$) can be found irrespective of the size $N$ of the state space (not shown), meaning that the relative speed-up between the optimal solution and LS grows with $N$ (cf. Fig. 2).

The optimal $\mathbf{S}_{\mathrm{opt}}$ induces a weight matrix $\mathbf{W}_{\mathrm{opt}}$ given by Eq. 9 and shown in Fig. 4A (middle). Notably, $\mathbf{W}_{\mathrm{opt}}$ is no longer symmetric, and its elements are much larger than in the Langevin symmetric solution $\mathbf{W}_{\mathrm{L}}$ with the same stationary covariance, albeit orders of magnitude smaller than in random networks of comparable decorrelation speed (Fig. 3C).

It is illuminating to visualize activity trajectories in the plane defined by the topmost and bottommost eigenvectors of $\mathbf{\Sigma}$, i.e. the first and last principal components (PCs) of the network activity (Fig. 4C). The distribution of interest is broad along some dimensions, and narrow along others. In order to sample efficiently, large steps ought to be taken along directions in which the distribution is broad, and small steps along directions in which the distribution is narrow. This is exactly what our optimal sampler does, whereas LS takes small steps along both broad and narrow directions (Fig. 4C).

## 6    Balanced E/I networks for fast sampling

We can further constrain our network to obey Dale's law, i.e. the separation of neurons into separate excitatory (E) and inhibitory (I) groups. The main difficulty in building such networks is that picking an arbitrary skew-symmetric matrix $\mathbf{S}$ in Eq. 9 will not yield the column sign structure of an E/I network in general. Therefore, we no longer have a parametric form for the solution matrix manifold on which to find the fastest network. However, by extending the methods of Sec. 5, described in

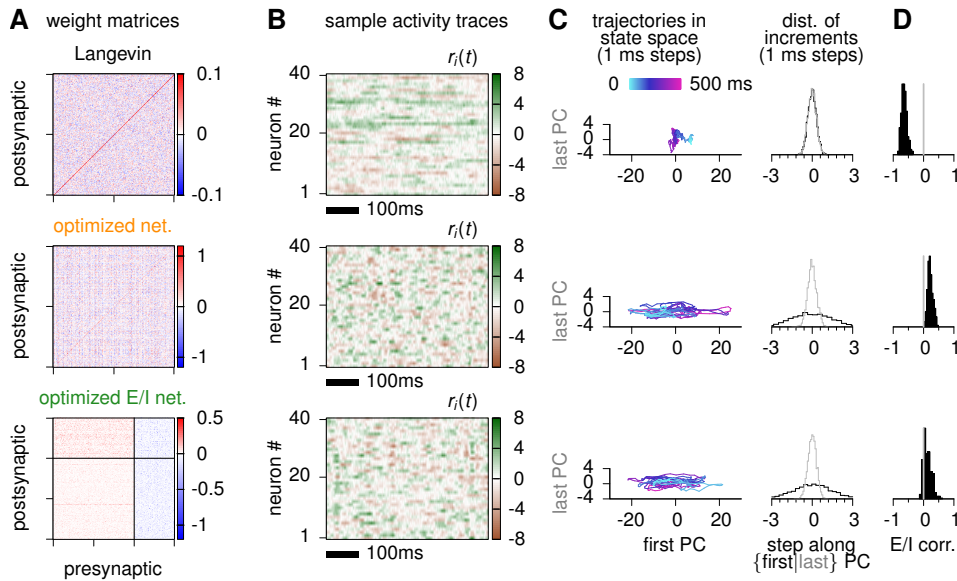

Figure 4: **Fast sampling with optimized networks. (A)** Synaptic weight matrices for the Langevin network (top), the fastest sampler (middle) and the fastest sampler that obeys Dale's law (bottom). Note that the synaptic weights in both optimized networks are an order of magnitude larger than in the symmetric Langevin solution. The first two networks are of size $N = 200$, while the optimized E/I network has size $N + N_I = 300$. **(B)** 500 ms of spontaneous network activity ($\mathbf{h} = 0$) in each of the three networks, for all of which the stationary distribution of $\mathbf{r}$ (restricted here to the first 40 neurons) is the same multivariate Gaussian. **(C)** Left: activity trajectories (the same 500 ms as shown in (B)) in the plane defined by the topmost and bottommost eigenvectors of the posterior covariance matrix $\mathbf{\Sigma}$ (corresponding to the first and last principal components of the activity fluctuations $\mathbf{r}(t)$). For the E/I network, the projection is restricted to the excitatory neurons. Right: distribution of increments along both axes, measured in 1 ms time steps. Langevin sampling takes steps of comparable size along all directions, while the optimized networks take much larger steps along the directions of large variance prescribed by the posterior. **(D)** Distributions of correlations between the time courses of total excitatory and inhibitory input in individual neurons.

detail in our SI, we can still formulate the problem as one of unconstrained optimization, and obtain the fastest, balanced E/I sampler.

We consider the posterior to be encoded in the activity of the $N = 200$ excitatory neurons, and add $N_I = 100$ inhibitory neurons which we regard as auxiliary variables, in the spirit of Hamiltonian Monte Carlo methods [11]. Consequently, the E-I and I-I covariances are free parameters, while the E-E covariance is given by the target posterior. For additional biological realism, we also forbid self-connections as they can be interpreted as a modification of the intrinsic membrane time constant of the single neurons, which in principle cannot be arbitrarily learned.

The speed optimization yields the connectivity matrix shown in Fig. 4A (bottom). Results for this network are presented in a similar format as before, in the same figures. Sampling is almost as fast as in the best (regularized) unconstrained network (compare yellow and green in Fig. 3), indicating that Dale's law – unlike the symmetry constraint implicitly present in Langevin sampling – is not fundamentally detrimental to mixing speed. Moreover, the network operates in a regime of excitation/inhibition balance, whereby the total E and I input time courses are correlated in single cells (Fig. 4D, bottom). This is true also in the unconstrained optimal sampler. In contrast, E and I inputs are strongly anti-correlated in LS.

# 7 Discussion

We have studied sampling for Bayesian inference in neural circuits, and observed that a linear stochastic network is able to sample from the posterior under a linear Gaussian latent variable model. Hidden variables are directly encoded in the activity of single neurons, and their joint activity undergoes moment-to-moment fluctuations that visit each portion of the state space at a frequency given by the target posterior density. To achieve this, external noise sources fed into the network are amplified by the recurrent circuitry, but preferentially amplified along the state-space directions of large posterior variance. Although, for the very simple linear Gaussian model we considered here, a purely feed-forward architecture would also trivially be able to provide independent samples (ie. provide samples that are decorrelated at the time scale of $\tau_\mathrm{m}$), the network required to achieve this is deeply biologically implausible (see SI).

We have shown that the choice of a *symmetric weight matrix* – equivalent to LS, a popular machine learning technique [2, 11, 12] that has been suggested to underlie neuronal network dynamics sampling continuous variables [6, 13] – is most unfortunate. We presented an analytical argument predicting dramatic slowing in high-dimensional latent spaces, supported by numerical simulations. Even in moderately large networks, samples were correlated on timescales much longer than the single-neuron decay time constant.

We have also shown that when the above symmetry constraint is relaxed, a family of other solutions opens up that can potentially lead to much faster sampling. We chose to explore this possibility from a normative viewpoint, optimizing the network connectivity directly for sampling speed. The fastest sampler turned out to be highly asymmetric and typically an order of magnitude faster than Langevin sampling. Notably, we also found that constraining each neuron to be either excitatory or inhibitory does not impair performance while giving a far more biologically plausible sampler. Dale's law could even provide a natural safeguard against reaching slow symmetric solutions such as Langevin sampling, which we saw was the worst-case scenario (cf. also SI).

It is worth noting that $\mathbf{W}_\mathrm{opt}$ is strongly nonnormal.[3] Deviation from normality has important consequences for the dynamics of our networks: it makes the network sensitive to perturbations along some directions in state space. Such perturbations are rapidly amplified into large, transient excursions along other, relevant directions. This phenomenon has been shown to explain some key features of spontaneous activity in primary visual cortex [9] and primary motor cortex [22].

Several aspects would need to be addressed before our proposal can crystalize into a more thorough understanding of the neural implementation of the sampling hypothesis. First, can local synaptic plasticity rules perform the optimization that we have approached from an algorithmic viewpoint? Second, what is the origin of the noise that we have hypothesized to come from external sources? Third, what kind of nonlinearity must be added in order to allow sampling from non-Gaussian distributions, whose shapes may have non-trivial dependencies on the observations? Also, does the main insight reached here – namely that fast samplers are to be found among nonsymmetric, nonnormal networks – carry over to the nonlinear case? As a proof of principle, in preliminary simulations, we have shown that speed optimization in a linearized version of a nonlinear network (with a tanh gain function) does yield fast sampling in the nonlinear regime, even when fluctuations are strong enough to trigger the nonlinearity and make the resulting sampled distribution non-Gaussian (details in SI).

Finally, we have also shown (see SI) that the Langevin solution is the only network that satisfies the detailed balance condition [23] in our model class; reversibility is violated in all other stochastic networks we have presented here (random, optimal, optimal E/I). The fact that these networks are faster samplers is in line with recent machine learning studies on how non-reversible Markov chains can mix faster than their reversible counterparts [24]. The construction of such Monte-Carlo algorithms has proven challenging [25, 26, 27], suggesting that the brain – if it does indeed use sampling-based representations – might have something yet to teach us about machine learning.

**Acknowledgements**  This work was supported by the Wellcome Trust (GH, ML), the Swiss National Science Foundation (GH) and the Gatsby Charitable Foundation (LA). Our code will be made freely available from GH's personal webpage.

## Footnotes

[1]"Nonnormal" should not be confused with "non-Gaussian": a matrix $\mathbf{M}$ is nonnormal iff $\mathbf{MM}^\top \neq \mathbf{M}^\top \mathbf{M}$.

[2]From a pure machine learning perspective, increasing $\sigma_\xi$ is not an option either: the increasing stiffness of Eq. 6 would either require the use of a very small integration step, or would lead to arbitrarily small acceptance ratios in the context of Metropolis-Hastings proposals.

[3]Indeed, the sum of the squared moduli of its eigenvalues accounts for only 25% of $\|\mathbf{W}_\mathrm{opt}\|_\mathrm{F}^2$ [7]. For a normal matrix $\mathbf{W}$ (such as the Langevin solution, $\mathbf{W}_\mathrm{L}$), $\sum_i |\lambda_i|^2 = \|\mathbf{W}\|_\mathrm{F}^2$, i.e. this ratio is 100%.

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
