[Supplementary Material]

# Fast Sampling-Based Inference in Balanced Neuronal Networks
# — Supplementary Information —

**Guillaume Hennequin**[1]
gjeh2@cam.ac.uk

**Laurence Aitchison**[2]
laurence@gatsby.ucl.ac.uk

**Máté Lengyel**[1]
m.lengyel@eng.cam.ac.uk

[1]Computational & Biological Learning Lab, Dept. of Engineering, University of Cambridge, UK
[2]Gatsby Computational Neuroscience Unit, University College London, UK

## 1 Lower bound on the maximum eigenvalue of the Langevin connectivity matrix

Here we give an informal derivation of the lower bound on $\mathrm{Re}\left(\lambda_{\max}^{\mathbf{W_L}-\mathbf{I}}\right)$ found in Eq. 7 of the main text. Let us recall that $(\mathbf{W_L} - \mathbf{I})$ is real and symmetric, so its eigenvalues are all real, and since $\mathbf{W_L} - \mathbf{I} = -\sigma_\xi^2 \mathbf{\Sigma}^{-1}$ we can write[1]

$$\lambda_{\max}^{\mathbf{W_L}-\mathbf{I}} \quad = \quad -\sigma_\xi^2 \lambda_{\min}^{\mathbf{\Sigma}^{-1}} \quad = \quad -\frac{\sigma_\xi^2}{\lambda_{\max}^{\mathbf{\Sigma}}} \tag{1}$$

Now, again because of its symmetry, $\mathbf{\Sigma}$ is a normal matrix, and so it is similar to (i.e. equal to the unitary transformation of) a diagonal matrix that contains its eigenvalues. Since unitary transformations preserve the Frobenius norm, we can write $\sum_{i,j} \Sigma_{ij}^2 = \sum_i \left(\lambda_i^{\mathbf{\Sigma}}\right)^2$ and since all the eigenvalues of $\mathbf{\Sigma}$ are positive, we have $N\left(\lambda_{\max}^{\mathbf{\Sigma}}\right)^2 \geq \sum_i \left(\lambda_i^{\mathbf{\Sigma}}\right)^2$. Plugging this into Eq. 1, we arrive at a bound that relates the maximum eigenvalue of $(\mathbf{W_L} - \mathbf{I})$ to a basic summary statistics, the sum of all (co)variances, of the posterior covariance matrix $\mathbf{\Sigma}$:

$$\lambda_{\max}^{\mathbf{W_L}-\mathbf{I}} \quad \geq \quad -\sigma_\xi^2 \sqrt{\frac{N}{\sum_{ij} \Sigma_{ij}^2}} \tag{2}$$

In the $N \to \infty$ limit, assuming that pairwise correlations do not vanish, the denominator is $\mathcal{O}(N^2)$, meaning that $0 > \lambda_{\max}^{\mathbf{W_L}-\mathbf{I}} \geq -\mathcal{O}(1/\sqrt{N})$: the slowest eigenmode of $\mathbf{W_L}$ becomes critically slow. To make this bound more concrete, let us assume that $\Sigma_{ii} \simeq \sigma_0^2$ (all posterior variances are roughly equal) and that the distribution of pairwise posterior correlations has zero mean and standard deviation $\sigma_r$. We can then rewrite Eq. 2 as

$$\lambda_{\max}^{\mathbf{W_L}-\mathbf{I}} \quad \geq \quad \frac{-(\sigma_\xi/\sigma_0)^2}{\sqrt{1 + N\sigma_r^2}} \tag{3}$$

which is Eq. 7 of our main text.

## 2 Minimization of the slowing cost $\psi_{\mathrm{slow}}$

Let us recall the definition of the slowing cost for convenience:

$$\psi_{\mathrm{slow}}(\mathbf{S}) = \frac{1}{2\tau_{\mathrm{m}} N^2} \int_0^\infty \left\| \mathbf{\Lambda}^{-\frac{1}{2}} \mathbf{K}(\mathbf{S}, \tau) \mathbf{\Lambda}^{-\frac{1}{2}} \right\|_{\mathrm{F}}^2 \, \mathrm{d}\tau \tag{4}$$

where $\mathbf{K}(\mathbf{S}, \tau) \equiv \left\langle \delta\mathbf{r}(t+\tau) \, \delta\mathbf{r}(t)^\top \right\rangle_t$.

From Ornstein-Uhlenbeck theory [1], we know that $\mathbf{K}(\mathbf{S}, \tau)$ obeys the following differential equation:

$$\tau_{\mathrm{m}} \frac{d\mathbf{K}(\mathbf{S}, \tau)}{d\tau} = [\mathbf{W}(\mathbf{S}) - \mathbf{I}] \, \mathbf{K}(\mathbf{S}, \tau) \tag{5}$$

such that for $\tau \geq 0$, we have $\mathbf{K}(\mathbf{S}, \tau) = e^{[\mathbf{W}(\mathbf{S})-\mathbf{I}] \, \tau/\tau_{\mathrm{m}}} \mathbf{\Sigma}$. We may thus rewrite $\psi_{\mathrm{slow}}(\mathbf{S})$ as

$$\psi_{\mathrm{slow}}(\mathbf{S}) = \frac{1}{2N^2} \mathrm{trace} \left[ \mathbf{V} \left( \int_0^\infty e^{\tau[\mathbf{W}(\mathbf{S})-\mathbf{I}]} \mathbf{U}\mathbf{U}^\top e^{\tau[\mathbf{W}(\mathbf{S})^\top - \mathbf{I}]} \mathrm{d}\tau \right) \mathbf{V}^\top \right] \tag{6}$$

with the shorthand notation $\mathbf{U} \equiv \mathbf{\Sigma}\mathbf{\Lambda}^{-\frac{1}{2}}$ and $\mathbf{V} \equiv \mathbf{\Lambda}^{-\frac{1}{2}}$. Equation 6 is the canonical form used in linear quadratic control theory [2] and affords the following gradient:

$$\frac{\partial \psi_{\mathrm{slow}}(\mathbf{S})}{\partial \mathbf{S}} = \frac{1}{N^2} \left[ (\mathbf{\Sigma}^{-1}\mathbf{P}\mathbf{Q})^\top - (\mathbf{\Sigma}^{-1}\mathbf{P}\mathbf{Q}) \right] \tag{7}$$

Here $\mathbf{P}$ and $\mathbf{Q}$ are the unique solutions of a pair of Lyapunov equations,

$$(\mathbf{W} - \mathbf{I})\mathbf{P} + \mathbf{P}(\mathbf{W} - \mathbf{I})^\top = -\mathbf{\Sigma}\mathbf{\Lambda}^{-1}\mathbf{\Sigma} \tag{8}$$

$$(\mathbf{W} - \mathbf{I})^\top \mathbf{Q} + \mathbf{Q}(\mathbf{W} - \mathbf{I}) = -\mathbf{\Lambda}^{-1}, \tag{9}$$

which can be solved efficiently [3], e.g. using the Matlab function `lyap`. Note also that $\psi_{\mathrm{slow}}(\mathbf{S}) = \mathrm{trace}(\mathbf{\Lambda}^{-1/2}\mathbf{P}\mathbf{\Lambda}^{-1/2})/2N^2$ [2].

The $L_2$-penalty term in the overal cost function (Eq. 13 of the main text) is more easily differentiated, yielding the gradient

$$\frac{\partial \mathcal{L}(\mathbf{S})}{\partial \mathbf{S}} = \frac{1}{N^2} \left[ (\mathbf{\Sigma}^{-1}\mathbf{P}\mathbf{Q})^\top - (\mathbf{\Sigma}^{-1}\mathbf{P}\mathbf{Q}) \right] + \frac{\lambda_{L_2}}{N^2} \left[ \mathbf{S}\mathbf{\Sigma}^{-2} + \mathbf{\Sigma}^{-2}\mathbf{S} \right] \tag{10}$$

which is skew-symmetric, as it should.

## 3 *Only* Langevin sampling (LS) satisfies detailed balance (in our model class)

Consider a Markov chain $\{\mathbf{x}_t\}$ with stationary distribution $p(\mathbf{x}_t)$ and a probability of transitioning from state $\mathbf{x}_t$ into state $\mathbf{x}_{t+1}$ given by $p(\mathbf{x}_{t+1}|\mathbf{x}_t)$. Detailed balance is satisfied if, and only if for any pair of states $(\mathbf{x}_t, \mathbf{x}_{t+1})$, we have

$$p(\mathbf{x}_{t+1}|\mathbf{x}_t) \, p(\mathbf{x}_t) = p(\mathbf{x}_t|\mathbf{x}_{t+1}) \, p(\mathbf{x}_{t+1}) \tag{11}$$

Equation 11 states that any state sequence $\mathbf{x}_t \rightarrow \mathbf{x}_{t+1}$ should be visited as often as the reverse sequence $\mathbf{x}_{t+1} \rightarrow \mathbf{x}_t$, that is, time is reversible. Taking the logarithm on both sides, we rewrite the detailed balance condition as

$$\log p(\mathbf{x}_{t+1}|\mathbf{x}_t) + \log p(\mathbf{x}_t) = \log p(\mathbf{x}_t|\mathbf{x}_{t+1}) + \log p(\mathbf{x}_{t+1}) \tag{12}$$

To see whether or not detailed balance holds in our samplers, we write the network dynamics (Eq. 3 in the main text) in discrete time, i.e. for $\epsilon \rightarrow 0$ we have

$$\mathbf{x}_{t+1} = \mathbf{x}_t + \epsilon \mathbf{A}\mathbf{x}_t + \sqrt{2\epsilon}\eta_t \tag{13}$$

where $\mathbf{A} \equiv \mathbf{W} - \mathbf{I} = (-\mathbf{I} + \mathbf{S})\mathbf{\Sigma}^{-1}$ is the effective connectivity (it includes the leak term), $\eta_t \sim \mathcal{N}(0, \mathbf{I})$, and both $\tau_{\mathrm{m}}$ and $\sigma_\xi$ have been set to unity without loss of generality.

Thus

$$\log p(\mathbf{x}_{t+1}|\mathbf{x}_t) = -\frac{N}{2} \log(2\pi\epsilon) - \frac{\|\mathbf{x}_{t+1} - (\mathbf{I} + \epsilon\mathbf{A})\mathbf{x}_t\|^2}{4\epsilon} \tag{14}$$

and, given that our samplers have the right stationary distribution $\mathcal{N}(0, \boldsymbol{\Sigma})$ (detailed balance is not required to show this, see [1]),

$$\log p(\mathbf{x}_t) = -\frac{N}{2} \log(2\pi) - \frac{1}{2} \log |\boldsymbol{\Sigma}| - \frac{1}{2} \mathbf{x}_t^\top \boldsymbol{\Sigma}^{-1} \mathbf{x}_t \tag{15}$$

Therefore, detailed balance is satisfied if, and only if for any state pair $(\mathbf{x}_t, \mathbf{x}_{t+1})$ we have:

$$\|\mathbf{x}_{t+1} - (\mathbf{I} + \epsilon \mathbf{A})\mathbf{x}_t\|^2 + 2\epsilon \mathbf{x}_t^\top \boldsymbol{\Sigma}^{-1} \mathbf{x}_t = \|\mathbf{x}_t - (\mathbf{I} + \epsilon \mathbf{A})\mathbf{x}_{t+1}\|^2 + 2\epsilon \mathbf{x}_{t+1}^\top \boldsymbol{\Sigma}^{-1} \mathbf{x}_{t+1} \tag{16}$$

Keeping only first-order terms in $\epsilon$, and inserting the parameterization $\mathbf{A} = (-\mathbf{I} + \mathbf{S})\boldsymbol{\Sigma}^{-1}$, we obtain the following necessary and sufficient condition for reversibility:

$$2(\mathbf{x}_{t+1} - \mathbf{x}_t)^\top \mathbf{S} \boldsymbol{\Sigma}^{-1} (\mathbf{x}_{t+1} - \mathbf{x}_t) = 0 \tag{17}$$

Clearly, if $\mathbf{S} = 0$, the condition is satisfied, therefore detailed balance holds. Conversely, if detailed balance holds, then Eq. 17 must hold for *any* pair $(\mathbf{x}_t, \mathbf{x}_{t+1})$, from which it is easy to see that $\mathbf{S} \boldsymbol{\Sigma}^{-1}$ must be zero, and therefore $\mathbf{S} = 0$ too. Therefore, only the Langevin solution (which corresponds to $\mathbf{S} = 0$) satisfies time reversibility.

## 4 LS is at the pessimum of the slowness cost function

Here we prove that LS corresponds to a pessimum of the slowness cost function $\psi_{\text{slow}}$ used throughout the paper to optimize for sampling speed. To do this, we show that the gradient $\partial \psi_{\text{slow}}/\partial \mathbf{S}$ is zero at $\mathbf{S} = 0$.

Let us assume that $\mathbf{S} = 0$. Then,

$$\mathbf{A} = -\sigma_\xi^2 \boldsymbol{\Sigma}^{-1} \tag{18}$$

such that the two Lyapunov equations (Eqs. 8 and 9) of Sec. 2 become:

$$\sigma_\xi^2 \left( \boldsymbol{\Sigma}^{-1} \mathbf{P} + \mathbf{P} \boldsymbol{\Sigma}^{-1} \right) = \boldsymbol{\Sigma} \boldsymbol{\Lambda}^{-1} \boldsymbol{\Sigma} \tag{19}$$

$$\sigma_\xi^2 \left( \boldsymbol{\Sigma}^{-1} \mathbf{Q} + \mathbf{Q} \boldsymbol{\Sigma}^{-1} \right) = \boldsymbol{\Lambda}^{-1} \tag{20}$$

Now,

$$\sigma_\xi^2 \left( \boldsymbol{\Sigma}^{-1} (\boldsymbol{\Sigma} \mathbf{Q} \boldsymbol{\Sigma}) + (\boldsymbol{\Sigma} \mathbf{Q} \boldsymbol{\Sigma}) \boldsymbol{\Sigma}^{-1} \right) = \sigma_\xi^2 \left( \mathbf{Q} \boldsymbol{\Sigma} + \boldsymbol{\Sigma} \mathbf{Q} \right) \tag{21}$$

$$= \sigma_\xi^2 \boldsymbol{\Sigma} \left( \boldsymbol{\Sigma}^{-1} \mathbf{Q} + \mathbf{Q} \boldsymbol{\Sigma}^{-1} \right) \boldsymbol{\Sigma} \tag{22}$$

$$= \boldsymbol{\Sigma} \boldsymbol{\Lambda}^{-1} \boldsymbol{\Sigma} \tag{23}$$

(the last equality uses Eq. 20). Thus, $\boldsymbol{\Sigma} \mathbf{Q} \boldsymbol{\Sigma}$ is a solution to Eq. 19, and since the solution is unique because $\mathbf{P}$ is positive definite [2], we conclude that

$$\mathbf{P} = \boldsymbol{\Sigma} \mathbf{Q} \boldsymbol{\Sigma} \tag{24}$$

i.e

$$\boldsymbol{\Sigma}^{-1} \mathbf{P} = \mathbf{Q} \boldsymbol{\Sigma} \tag{25}$$

Using this result in Eq. 7, together with the fact that $\mathbf{P}$, $\mathbf{Q}$ and $\boldsymbol{\Sigma}$ are symmetric, we compute:

$$\frac{\partial \psi_{\text{slow}}(\mathbf{S})}{\partial \mathbf{S}} \propto \left( \boldsymbol{\Sigma}^{-1} \mathbf{P} \mathbf{Q} \right)^\top - \boldsymbol{\Sigma}^{-1} \mathbf{P} \mathbf{Q} \tag{26}$$

$$= \mathbf{Q} \mathbf{P} \boldsymbol{\Sigma}^{-1} - \mathbf{Q} \boldsymbol{\Sigma} \mathbf{Q} \tag{27}$$

$$= \mathbf{Q} (\boldsymbol{\Sigma}^{-1} \mathbf{P})^\top - \mathbf{Q} \boldsymbol{\Sigma} \mathbf{Q} \tag{28}$$

$$= \mathbf{Q} (\mathbf{Q} \boldsymbol{\Sigma})^\top - \mathbf{Q} \boldsymbol{\Sigma} \mathbf{Q} \tag{29}$$

$$= \mathbf{Q} \boldsymbol{\Sigma} \mathbf{Q} - \mathbf{Q} \boldsymbol{\Sigma} \mathbf{Q} \tag{30}$$

$$= 0 \tag{31}$$

At this stage, we have shown that $\mathbf{S} = 0$ corresponds to a critical point of $\psi_{\text{slow}}$. The fact that small, random (unstructured) perturbations of $\mathbf{S}$ around 0 only decrease $\psi_{\text{slow}}$ (Fig. 3 of the main text) suggest that LS is in fact (locally, but perhaps also globally) the slowest possible sampler for our problem.

## 5 Details of the balanced E/I network optimization

To build optimized networks that obey Dale's law, we assume that there are $N_{\text{exc.}} = N$ excitatory neurons, where $N$ is the dimension of the distribution we want to sample from, and $N_{\text{inh.}}$ inhibitory neurons whose activity distribution is irrelevant (i.e. we regard inhibitory neurons as auxiliary sampling variables, in the spirit of Hamiltonian Monte Carlo methods [4]). In the main paper, $N = 200$ and $N_I = 100$. Let $M = N_{\text{exc.}} + N_{\text{inh.}}$ denote the total network size. The dynamics do not change, i.e. we still have

$$\text{d}\mathbf{r} = \frac{\text{d}t}{\tau_{\text{m}}} \left[ -\mathbf{r}(t) + \mathbf{W}\mathbf{r}(t) + \mathbf{F}\mathbf{h} \right] + \sigma_\xi \sqrt{\frac{2}{\tau_{\text{m}}}} \, \text{d}\boldsymbol{\xi}(t) \tag{32}$$

The connectivity matrix $\mathbf{W}$ is now made of $N$ positive columns followed by $N_I$ negative columns. This makes it difficult to apply the approach of Sec. 5 of the main text, as picking an arbitrary skew-symmetric matrix $\mathbf{S}$ in Eq. 11 (main text) will not yield the column sign structure of an E/I network in general. Therefore, we no longer have a parametric form for the solution matrix manifold on which to search for the fastest network. However, with a few simple variations, we can still formulate the problem as one of unconstrained optimization, as explained now.

The first step is to enforce Dale's law through the following re-parameterization of $\mathbf{W}$:

$$W_{ij} = (1 - \delta_{ij}) \, s_j \, \exp \beta_{ij} \tag{33}$$

where $s_j$ is a fixed sign that depends only on presynaptic neuron $j$ ($s_j = +1$ for $j \leq N$, $-1$ otherwise), and the $\beta_{ij}$'s are unconstrained free parameters. Note that we do not allow for autapses, hence the $(1 - \delta_{ij})$ term in Eq. 33). Second, since the target posterior distribution specifies only the $N \times N$ upper-left quadrant $\boldsymbol{\Sigma}$ of the overall covariance matrix which we denote by $\boldsymbol{\Sigma}_{\text{tot}}$, we are free to optimize over the other quadrants. We parameterize $\boldsymbol{\Sigma}_{\text{tot}}$ by its Cholesky factor:

$$\boldsymbol{\Sigma}_{\text{tot}} = \mathbf{L}\mathbf{L}^\top, \qquad \mathbf{L} \equiv \begin{pmatrix} \mathbf{L}_{11} & 0 \\ \mathbf{L}_{12} & \mathbf{L}_{22} \end{pmatrix} \tag{34}$$

where $\mathbf{L}_{11}$ is the Cholesky factor of the posterior covariance matrix $\boldsymbol{\Sigma}$ (i.e. $\boldsymbol{\Sigma} = \mathbf{L}_{11}\mathbf{L}_{11}^\top$), and the two matrices $\mathbf{L}_{12}$ and $\mathbf{L}_{22}$ are free parameters. Note that $\mathbf{L}_{12}$ is a full rectangular matrix of size $N_I \times N$, while $\mathbf{L}_{22}$ is lower-triangular with dimensions $N_I \times N_I$. Third, in order to force the network to sample from the right multivariate Gaussian distribution, we incorporate the Lyapunov equation (cf. Eq. 4 in the main text) as an additional constraint in our loss function. This additional term reads:

$$\psi_{\text{sol.}} \equiv \frac{1}{2M^2} \left\| (\mathbf{W} - \mathbf{I})\boldsymbol{\Sigma}_{\text{tot}} + \boldsymbol{\Sigma}_{\text{tot}}(\mathbf{W} - \mathbf{I})^\top + 2\sigma_\xi^2 \mathbf{I} \right\|_{\text{F}}^2 \tag{35}$$

When $\psi_{\text{sol.}}$ is zero, the Lyapunov equation (Eq. 4 in the main text) is satisfied, and therefore the stationary covariance matrix of the network dynamics matches $\boldsymbol{\Sigma}_{\text{tot}}$. In particular, their upper-left quadrant would then be equal, meaning that the excitatory sub-network would be sampling from the right posterior.

Note that $\psi_{\text{sol.}}$ depends on both the $\beta_{ij}$'s (through $\mathbf{W}$) and the free covariance parameters $\mathbf{L}_{12}$ and $\mathbf{L}_{22}$ (through $\boldsymbol{\Sigma}_{\text{tot}}$). The corresponding gradients can be obtained after a bit of algebra, and read:

$$\frac{\partial \psi_{\text{sol.}}}{\partial \mathbf{L}} = \frac{2}{M^2} \left[ (\mathbf{G}\mathbf{A}) + (\mathbf{G}\mathbf{A})^\top \right] \mathbf{L} \tag{36}$$

$$\frac{\partial \psi_{\text{sol.}}}{\partial \mathbf{W}} = \frac{2}{M^2} \mathbf{G}\boldsymbol{\Sigma}_{\text{tot}} \tag{37}$$

where

$$\mathbf{G} \equiv (\mathbf{W} - \mathbf{I})\boldsymbol{\Sigma}_{\text{tot}} + \boldsymbol{\Sigma}_{\text{tot}}(\mathbf{W} - \mathbf{I})^\top + 2\sigma_\xi^2 \mathbf{I} \tag{38}$$

Note that we are interested only in the lower-triangular part of Eq. 36. The application of the chain rule to go from $\mathbf{W}$ to the $\beta_{ij}$'s in Eq. 37 is straightforward (it can be performed element-wise).

The total cost function we minimize is

$$\mathcal{L} \equiv \psi_{\text{sol.}} + \lambda_{\text{slow}}\psi_{\text{slow}} + \frac{\lambda_{L_2}}{2M^2} \|\mathbf{W}\|_{\text{F}}^2 \tag{39}$$

where $\psi_{\text{slow}}$ penalizes the magnitude of lagged auto- and cross-correlations over an infinite time horizon, in the excitatory sub-network only. To give a formal definition of $\psi_{\text{slow}}$, let us use the

notation $\tilde{\mathbf{A}}$ to denote the zeroing of all elements but those in the upper-left $N \times N$ quadrant of any $M \times M$ matrix $\mathbf{A}$. The modified slowness loss is then written as

$$\psi_{\text{slow}}(\mathbf{S}) \equiv \frac{1}{2\tau_{\mathrm{m}} N^2} \int_0^\infty \left\| \mathbf{\Lambda}^{-\frac{1}{2}} \tilde{\mathbf{K}}(\mathbf{S}, \tau) \mathbf{\Lambda}^{-\frac{1}{2}} \right\|_{\mathrm{F}}^2 \, \mathrm{d}\tau \tag{40}$$

where $\mathbf{\Lambda}$ is a diagonal matrix that contains the diagonal of $\mathbf{\Sigma}$ in its upper-left quadrant and zeros everywhere else. The kernel $\mathbf{K}$ is defined as in the main text, i.e.

$$\mathbf{K}(\mathbf{S}, \tau) \quad \equiv \quad \langle \mathbf{r}(t) \mathbf{r}(t + \tau)^\top \rangle_t \tag{41}$$

$$= \quad e^{(\mathbf{W} - \mathbf{I}) \tau / \tau_{\mathrm{m}}} \mathbf{\Sigma}_{\text{tot}} \tag{42}$$

(the second equality can be found in e.g. [1]). Observing that $\tilde{\mathbf{K}}(\mathbf{S}, \tau) = \tilde{\mathbf{I}} \mathbf{K}(\mathbf{S}, \tau) \tilde{\mathbf{I}}$, and making the change of variable $\tau / \tau_{\mathrm{m}} \to \tau$, we can rewrite the slowness cost as

$$\psi_{\text{slow}}(\mathbf{S}) = \frac{1}{2N^2} \operatorname{trace} \left[ \mathbf{V} \left( \int_0^\infty e^{\tau(\mathbf{W} - \mathbf{I})} \mathbf{U} \mathbf{U}^\top e^{\tau(\mathbf{W} - \mathbf{I})^\top} \mathrm{d}\tau \right) \mathbf{V}^\top \right] \tag{43}$$

with

$$\mathbf{V} \equiv \mathbf{\Lambda}^{-\frac{1}{2}} \tilde{\mathbf{I}} \qquad\qquad = \quad \tilde{\mathbf{\Lambda}}^{-\frac{1}{2}} \tag{44}$$

$$\mathbf{U} \equiv \mathbf{\Sigma} \tilde{\mathbf{I}} \mathbf{\Lambda}^{-\frac{1}{2}} \qquad\qquad = \quad \mathbf{\Sigma} \tilde{\mathbf{\Lambda}}^{-\frac{1}{2}} \tag{45}$$

As in the main text we have to solve two dual Lyapunov equations for matrices $\mathbf{P}$ and $\mathbf{Q}$:

$$\mathbf{A}\mathbf{P} + \mathbf{P}\mathbf{A}^\top = -\mathbf{\Sigma}_{\text{tot}} \tilde{\mathbf{\Lambda}}^{-1} \mathbf{\Sigma}_{\text{tot}} \tag{46}$$

$$\mathbf{A}^\top \mathbf{Q} + \mathbf{Q}\mathbf{A} = -\tilde{\mathbf{\Lambda}}^{-1} \tag{47}$$

Thus [2],

$$\psi_{\text{slow}} = \frac{1}{2N^2} \operatorname{trace} \left( \tilde{\mathbf{\Lambda}}^{-\frac{1}{2}} \mathbf{P} \tilde{\mathbf{\Lambda}}^{-\frac{1}{2}} \right) \tag{48}$$

and the gradient w.r.t. the synaptic weights is again given by

$$\frac{\partial \psi_{\text{slow}}}{\partial \mathbf{W}} = \frac{\mathbf{Q}\mathbf{P}}{N^2} \tag{49}$$

The gradient w.r.t. the Cholesky factor $\mathbf{L}$ requires a bit more algebra, and reads

$$\frac{\partial \psi_{\text{slow}}}{\partial \mathbf{L}} = 2 \left[ \left( \tilde{\mathbf{\Lambda}}^{-1} \mathbf{\Sigma}_{\text{tot}} \mathbf{Q} \right) + \left( \tilde{\mathbf{\Lambda}}^{-1} \mathbf{\Sigma}_{\text{tot}} \mathbf{Q} \right) \right] \mathbf{L} \tag{50}$$

We used again the L-BFGS algorithm (from the NLopt library) to optimize $\mathcal{L}$ (Eq. 39), with parameters $\lambda_{L_2} = \lambda_{\text{slow}} = 0.1$.

## 6 Connection to Newton preconditioning

Given an arbitrary target distribution $p(\mathbf{r}) \propto \exp[-V(\mathbf{r})]$, LS can be more generally written as [5]

$$\mathrm{d}\mathbf{r} = -\mathbf{B}\nabla_{\mathbf{r}} V(\mathbf{r}(t)) \mathrm{d}t + \sqrt{2\mathbf{B}} \mathrm{d}\boldsymbol{\xi}(t) \tag{51}$$

where $\boldsymbol{\xi}$ is a unitary Wiener process and $\mathbf{B}$ is an arbitrary positive definite matrix (a "preconditioning" matrix). Note that the noise must also be preconditioned.

Inspired by classical Newton methods for gradient-based optimization, which take into account the curvature of the objective function, Martin et al. [5] advocate the use of the inverse Hessian as a preconditioner, shown to yield substantial speed improvements in many cases. In our Gaussian case, picking the inverse Hessian $\mathbf{H}^{-1} = -\mathbf{\Sigma}$ of the log-posterior as a preconditioner gives exactly:

$$\mathrm{d}\mathbf{r}(t) = -\mathbf{r}(t) + \sqrt{2\mathbf{\Sigma}} \mathrm{d}\boldsymbol{\xi}(t) \tag{52}$$

i.e. it pushes the target covariance in the noise term and removes all recurrent interactions (note that we have assumed zero posterior mean without loss of generality).

Our optimized samplers are thus quite different from smartly preconditioned Langevin sampling, in that they can still make use of *independent* noise sources.

# 7  Why feed-forward networks are insufficient

In the main text, we have shown how recurrent neural networks can sample fast from multivariate Gaussian distributions. Algorithmically, it is straightforward to draw independent samples from such distributions, and indeed, the standard Cholesky sampling algorithm can be seen as a feed-forward neural network in which the input layer is made of independent noisy neurons, and the output layer (consisting of the neurons that represent the posterior) mixes those inputs, as well as the external stimulus, linearly. However, this solution assumes that some neurons are stochastic and uncorrelated, and their only role is generating noise for the rest of the brain, while others (those representing the posterior in the output layer) respond deterministically to their inputs. We find this dichotomy physiologically highly unrealistic. Moreover, a feedforward layout is inconsistent with the ubiquitously recurrent nature of cortical connectivity, of which, in contrast, our networks make optimal use to support the computation.

# 8  Application to a nonlinear, non-Gaussian system

A fundamental motivation for focusing on recurrent networks is the intuition that only such complex architectures will be able to sample from non-Gaussian posterior distributions in which not only the mean, but also higher-order moments might have non-trivial dependencies on the input. Although the analytical results of the main text, in particular the gradient of the slowness cost function, are limited to linear networks, we have made preliminary explorations of nonlinear, non-Gaussian systems through simulations.

Our starting point is a nonlinear, random network with stochastic dynamics of the form:

$$d\mathbf{u} = \frac{dt}{\tau_{\mathrm{m}}}\left[-\mathbf{u}(t) + \mathbf{W}^{(0)}\mathbf{r}(t)\right] + \sigma_\xi\sqrt{\frac{2}{\tau_{\mathrm{m}}}}d\xi(t) \qquad \text{with} \quad r_i(t) = \tanh[u_i(t)] \qquad (53)$$

where $W_{ij}^{(0)} \sim \mathcal{N}(0, R^2/N)$ such that $\mathbf{W}^{(0)}$ has a circular eigenvalue spectrum of radius $R = 5$. We set the network size to $N = 200$. A sample activity trace from this nonlinear system, $r_i(t)$, is shown in Fig. 1A (blue). Neurons tend to spend prolonged periods of time at either lower or upper saturation of their nonlinear gain function ($\tanh$), yielding an autocorrelation length about 3-4 times greater than the membrane time constant (Fig. 1B, blue). There are strong negative and positive correlations in the joint activity of neuron pairs (Fig. 1C, x-axis).

We then asked if we could build a nonlinear network that would sample from (approximately) the same distribution, but faster. To apply our linear framework, we first estimated the covariance matrix $\mathbf{\Sigma_u} \equiv \langle \mathbf{u}(t)\mathbf{u}(t)^\top \rangle_t$ on the basis of a 100 second-long simulation of the nonlinear stochastic dynamics of Eq. 53. We then built an optimal linear network to sample from a normal distribution with covariance $\mathbf{\Sigma_u}$ as described in the main text, and finally used the resulting connectivity matrix $\mathbf{W}$ in place of $\mathbf{W}^{(0)}$ in the dynamics of Eq. 53.

This procedure yielded a nonlinear network that turned out to sample from approximately the same non-Gaussian multivariate distribution $p(\mathbf{r})$ as the original nonlinear network. Indeed, individual pairwise correlations $\langle r_i(t)r_j(t)\rangle_t$ approximated those in the original network to a good degree (Fig. 1C), as did individual marginals of $p(r_i)$ (not shown). Importantly, sampling was several times faster, as can be inferred from the sample activity trace of Fig. 1A (red) and as summarized in the average autocorrelation of $r_i(t)$ shown in Fig. 1B (red). Note, that since the original nonlinear network often operates close to saturation, and thus makes heavy use of its nonlinearities, it is not at all trivial that our speed optimization based on a linear approximation work so well. In fact, if, instead of the speed optimized nonnormal weight matrix, we use the corresponding Langevin solution from the linearized dynamics, then the nonlinear version of the dynamics does not only slow down but even fails to match the correct stationary distribution. This is because the Langevin solution encodes the posterior distribution in the principal eigenvectors of the weight matrix, and those are the directions along which saturation occurs the most.

Figure 1: Speeding up sampling for a non-Gaussian distribution in a nonlinear system. **(A)** Sample activity traces for a single unit in the nonlinear, chaotic network (blue), and in the optimized network (red). **(B)** Firing rate autocorrelation $\langle r_i(t)r_i(t + k\tau_{\mathrm{m}})\rangle_t$ (with $r_i$ transformed to z-score) averaged across neurons (flanking lines denote $\pm$ one std.), for the two networks. **(C)** Pairwise activity correlations $\langle r_i(t)r_j(t)\rangle_t$ (with the $r_k$'s transformed to z-scores) in the nonlinear, chaotic network (x-axis) vs. those in the optimized network (y-axis).

## Footnotes

[1]For a non-singular matrix $\mathbf{M}$, the eigenvalues of $\mathbf{M}^{-1}$ are the inverses of those of $\mathbf{M}$; and since $\mathbf{\Sigma}$ is a positive definite covariance matrix, all its eigenvalues are positive, which yields Eq. 1.