[Reviews · NeurIPS 2014]

Submitted by Assigned_Reviewer_14

Summary: This paper deals with sampling methods based on linear rate-based neural-networks. First, it shows that symmetric weights (a common constraint in many models) significantly hurt the mixing rate. Then it shows that a (more physiological) non-normal network can have a much faster mixing rate, if the connectivity is optimized for this purpose. This works even if more biological constraints (Dale's law) are imposed.

Clarity: Very clear and well organized paper.

Originality: To the best of my knowledge, these results are novel.

Significance: This is not completely clear too me. The importance of this paper mainly depends on how does this generalizes to non-linear networks - which are more physiological, and can be used to generate more sophisticated samplers. Perhaps this could be shown in a simulation. If this does not generalize, then the significance critically depends on how the issues below are resolved.

Quality: The derivations appear to be correct and elegant. However, I'm quite confused as to what is the advantage of the suggested optimized networks. Given a linear network as defined in the paper, it is straightforward to construct a network that mixes as fast as possible (with timescale \tau_m) - if we are allowed to use more neurons in the network than components in the sampled vector (r). Simply have two layers (r & h) of neurons with no connections within each layer, and a feedforward connectivity \sqrt(\Sigma-I) from the 'hidden' layer h to the visible layer r. Then it straightforward to show that we can sample from the r layer with the timescale of a single neuron. Clearly, this is similar to the pre-conditioned solution, except the noise sources are not external. Note that you can make it more physiological by adding recurrent connections to the hidden layer (and modifying accordingly the feedforward connectivity to \sqrt(\Sigma-I)*A, where A is the appropriate 'whitening' matrix) and outgoing connections from the r layer (to some other, external neurons). Also we can maintain Dale's Law by summing together several hidden layers. I'm not sure what is the problem with such a simple solution. My final assessment of this paper will mainly depend on the answer to this question, as well the following (closely related) questions:

1) What is the asymptotic scaling of slowing factor (\tau_max / \tau_m) for the optimized networks? Can you always maintain a single neuron timescale for the mixing, as N increases? Or do you have to decrease the simulation timestep in order to maintain numerical stability? If so, by how much?

2) Why do you need the L2-norm penalty on Eq. 13? Is it to maintain numerical stability? Does the network has to be close the edge of stability in order to mix fast?

*** After Author's feedback ***
The author feed back have mostly satisfied my objections. I'm still not convinced that my suggestion is not biophysically plausible. If the hidden neurons have recurrent connections, which are then "compensated for" by the feedforward connectivity to the visible neurons, it is not necessarily wasteful, since the hidden neurons can simultaneously do something else (e.g., not sampling). In other words, using feedforward connectivity, you can use any other part of the brain as a stochastic "source" which generates fast sampling. It is not clear to me how this would be non-consistent with literature. Anyway, if you can show that your method works also for non-linear networks, this makes this issue less important.
Summary: Though this paper is nicely written and executed.

Submitted by Assigned_Reviewer_27

The paper discusses a neuro-dynamical framework that achieves a fast sampling with a posterior distribution.
The author(s) analyzed the mixing of Langevin sampling (LS) and derived a function that controls the mixing speed. LS is indeed slow by since the function is large. The author(s) proposed to minimize the function with a regularization term and derived a fast sampling method, where the connections between neurons are no longer symmetric. The author(s) confirmed their theory by computer simulations, with a more biologically feasible case.

[quality]
Although the model discussed in the paper is too simple (a linear Gaussian latent variable model), the analysis that combined a stochastic differential equation and a control theory seems interesting and novel enough.

[clarity]
The paper is well written and rather easy to follow.

[originality]
Not clear.

[significance]
The significance of the results on neuroscience is not clear.
Summary: Although the model discussed is simple, the analysis seems interesting and novel enough. However, the significance of the results on neuroscience is not clear.

Submitted by Assigned_Reviewer_35

Summary: The author proposed a new sampling method from the appropriate posterior distributions in the brain. Conventional techniques of sampling such as Markov chain Monte Carlo (MCMC) algorithm are very slow. The authors tried to overcome this difficulty by considering the well-known Langevin sampling (LS) recipe and analyzing their propertied Based on these consideration, they proposed a neural network model that is optimally fast, and hence orders of magnitude faster than LS.

This paper was systematically written and well organized. Thus, it is not so difficult for wide areas of readers to understand this paper . They started from a linear Gaussian latent model, and consider linear recurrent stochastic dynamics for Langevin sampling (LS). They mathematically analyzed statistical properties by equations (8)-(10) in section 3, and the general solution and sampling speed by equations (11) and (12) in section 4. The presented results are excellent. Based on these theoretical considerations, they succeeded in obtaining the optimal sampler which is an order of magnitude faster than either Langevin or Gibbs sampling in section 5. Finally, they constraint the neural network model to obey Dale’s law and provide balanced excitatory-inhibitory (E/I) network for fast sampling. This proposal is one of fruitful candidates for functional role of the balanced E/I network.

Based on these my opinions, I think that this paper has broad interest for both neuroscience and machine learning communities. I recommend that this paper is suitable for presentation in NIPS.

Summary: II think that this paper has broad interest for both neuroscience and machine learning communities. I recommend that this paper is suitable for presentation in NIPS.
Author Feedback
Author rebuttal: We thank all three reviewers for their time and thorough evaluation of our paper.

Reviewer 14

1. Nonlinear cases. We agree that the generalization to nonlinear cases is an important extension. In fact, we have already started developing a mathematical framework to do just that, and our preliminary results suggest that the main insight of our linear analysis provides - namely that good solutions (fast network dynamics) are to be looked for among nonsymmetric, nonnormal networks - carries over to nonlinear networks. As a proof of principle, we have shown that optimizing a linearized version of a nonlinear network for speed does yield fast sampling in the nonlinear regime (with the fluctuations being strong enough that the [tanh] nonlinearity is effective). Moreover, the transient nature of amplification in the optimized nonlinear network is such that pairwise correlations are practically unaffected compared to the linear network. This was not the case in a nonlinear version of the symmetric Langevin solution. Space constraints won't allow us to add those results to the main text, but we will outline them briefly in the discussion and include a figure in the SM.

2. Overall significance. We emphasize that the purpose of our paper was not to try and find the fastest sampler in the space of all linear networks, but to find it in the space of plausible cortical-like architectures. We will rewrite part of our introduction to make that point even clearer. Cortical connectivity has a deeply recurrent nature which is thought to support complex computations, and our goal was to elucidate how recurrent connections can be exploited to support fast sampling. From a machine learning viewpoint, our paper does not provide a better method for sampling multivariate Gaussians by any means. Nevertheless, we believe our analysis of the simple Gaussian case does provide important insights about what networks can be expected to provide fast sampling in more complex scenarios (cf. point 1 above). We also emphasize that the technical contribution our paper, the combination of stochastic methods and control theory - which our paper is the first to provide to our knowledge -, has the potential to inspire new developments in sampling algorithms in machine learning, and evidently makes our work non-incremental from a methodological perspective as well. We will highlight these points better in the introduction and discussion.

As for the reviewer's suggestion specifically: if we understood it correctly, the simple sampler suggested there corresponds to the well-known Cholesky method commonly used to sample multivariate Gaussians, which is of course the method of choice in this case if only algorithmic considerations matter. However, neurally, this implies that only some neurons are stochastic and unconnected, while others (in the r layer) are not, which is clearly physiologically unrealistic. Adding recurrent connections to the solution as suggested, just for the sake of making the network look more realistic, also seems unsatisfying: those connections would not be advantageous in any way, in fact they would need to be "compensated for". In the class of networks we consider, the recurrent connections are those that actively support the computation, by both shaping the independent noise components into the right covariance, and ensuring sampling speed. This is consistent with a vast literature on experience-dependent synaptic plasticity of recurrent connections within cortex. We will discuss this briefly in the paper.

Finally, the need to ultimately understand fast sampling in nonlinear networks is in fact a key motivation for focusing on recurrent networks, and for abandoning simple feedforward solutions. We will clarify this in the introduction.

3. The slowing factor for the optimal network does not depend on N (as observed empirically in simulations with networks of size 10 to 500). Moreover, the magnitude (spectral norm) of the optimal W is also O(1). Individual weights are consistently found to be of order g/\sqrt{N} where g is a large constant (~10). For a given N, integration of a nonnormal differential equation requires more care compared to a normal one, but the time step need not decrease as N grows. We will add this note to our section 5.

4. The cost function decreases very fast as the weights move along the optimal direction, but then becomes much flatter, though it still decreases. Therefore, optimization can result in synaptic weights growing very large, at almost no measurable benefit past some threshold. Given that the brain has energy constraints (often measured as the L2 norm of synaptic weights), it made sense to include a regularizer to avoid the formation of unnecessarily (and perhaps unphysiologically) strong synapses. For numerical stability, the reviewer is right that keeping synapses as weak as possible is helpful -- but that's just a happy coincidence here: as it happens, our cost function can be near-minimized with reasonably small synaptic weights (cf. also answer 3 above).

A further clarification: our optimal networks aren't technically close to the edge of stability (the Langevin solution is, which is precisely what makes it slow!). Our networks are sensitive to transient perturbations along some directions in state space, but asymptotically they are deeply stable, which is what makes them fast. We will make that clearer in the text.

Reviewer 27

Overall significance of our work to neuroscience: see our point 2 for reviewer 14 above.

Originality: we emphasize that our analytical characterization of slowing in the standard Langevin sampling is completely novel, and the use of control-theoretic methods (and indeed, speed optimization in the first place) to tackle slowing is unprecedented to our knowledge.

Reviewer 35

Thank you for your positive evaluation of our work.